# Global and gene-specific translational regulation in *Escherichia coli* across different conditions

**Di Zhang**[1,2‡], **Sophia Hsin-Jung Li**[3,4,5‡], **Christopher G. King**[3], **Ned S. Wingreen**[3,6]*, **Zemer Gitai**[3]*, **Zhiyuan Li**[1,2]*

**1** Center for Quantitative Biology, Academy for Advanced Interdisciplinary Studies, Peking University, Beijing, China, **2** Peking-Tsinghua Center for Life Sciences, Academy for Advanced Interdisciplinary Studies, Peking University, Beijing, China, **3** Department of Molecular Biology, Princeton University, Princeton, New Jersey, United States of America, **4** Institute of Bioengineering, School of Life Sciences, Swiss Federal Institute of Technology Lausanne, Lausanne, Switzerland, **5** Global Health Institute, School of Life Sciences, Swiss Federal Institute of Technology Lausanne, Lausanne, Switzerland, **6** Lewis-Sigler Institute for Integrative Genomics, Princeton University, Princeton, New Jersey, United States of America

‡ These authors are joint first authors on this work.
* wingreen@princeton.edu (NSW); zgitai@princeton.edu (ZG); zhiyuanli@pku.edu.cn (ZL)

**Data Availability Statement:** The RNA-seq and ribosome profiling data have been publicly published and are available in the GEO database with accession number GSE182100: https://www.

## Abstract

How well mRNA transcript levels represent protein abundances has been a controversial issue. Particularly across different environments, correlations between mRNA and protein exhibit remarkable variability from gene to gene. Translational regulation is likely to be one of the key factors contributing to mismatches between mRNA level and protein abundance in bacteria. Here, we quantified genome-wide transcriptome and relative translation efficiency (RTE) under 12 different conditions in *Escherichia coli*. By quantifying the mRNA-RTE correlation both across genes and across conditions, we uncovered a diversity of gene-specific translational regulations, cooperating with transcriptional regulations, in response to carbon (C), nitrogen (N), and phosphate (P) limitations. Intriguingly, we found that many genes regulating translation are themselves subject to translational regulation, suggesting possible feedbacks. Furthermore, a random forest model suggests that codon usage partially predicts a gene's cross-condition variability in translation efficiency; such cross-condition variability tends to be an inherent quality of a gene, independent of the specific nutrient limitations. These findings broaden the understanding of translational regulation under different environments and provide novel strategies for the control of translation in synthetic biology. In addition, our data offers a resource for future multi-omics studies.

## Author summary

The central dogma connects DNA, RNA, and protein through transcription and translation. However, with the development of transcriptome and proteomics technology, it has been widely reported that mRNA abundance is not a comprehensive indicator of protein abundance. Translational regulation is critical in resolving this type of mismatch. It has

ncbi.nlm.nih.gov/geo/query/acc.cgi?acc=
GSE182100.

**Funding:** This work was supported by the National Natural Science Foundation of China (grant number: 32071255 to ZYL); U.S. Department of Health and Human Services, National Institutes of Health (grant number: R01 GM082938 to NSW, DP1AI124669 to ZG); and National Science Foundation through the Center for the Physics of Biological Function (PHY-1734030 to NSW). Funding for open access charge: National Natural Science Foundation of China. The funders had no role in study design, data collection and analysis, decision to publish, or preparation of the manuscript.

**Competing interests:** The authors have declared that no competing interests exist.

been reported that bacteria respond to heat stress, oxidative stress, and other stressful environments through translational regulation. Nutrient limitations are also fundamental challenges for bacteria, with many unknowns in their adaptation strategies. Using transcriptome and translatome quantification, we uncovered a diversity of gene-specific translational regulations, cooperating with transcriptional regulations, in response to carbon (C), nitrogen (N), and phosphate (P) limitations. Furthermore, we found that codon bias contributes substantially to gene-specific translational regulation. Our findings broaden the understanding of translational regulation under environmental changes and may assist in the design of effective translation strategies in synthetic biology.

## Introduction

The central dogma connects DNA, RNA, and protein through transcription and translation. While transcriptional regulation has been extensively studied in the past century [1], how well transcript level represents protein abundance remains controversial [2,3]. Despite the overall positive correlation between mRNA and protein abundance when different genes are compared in bacteria, genes with similar mRNA abundance may show large differences in protein abundance [4]. Given that a non-negligible portion of the bacterial genome exists as polycistrons, the difference in protein abundance can be traced to widespread differences in translational capacity between genes, e. g. arising from mRNA secondary structure, codon usage bias, ribosome binding sites, riboswitches, and leader peptides [5–9]. Not only do these factors result in differential protein synthesis under steady-state conditions, many of them also respond to external stressors, confirming that translation is also a crucial stage in gene expression regulation. For example, the hairpin structure in the 5'UTR of *pfrA* in *Listeria* opens at high temperatures to facilitate translation [10].

Cells regulate their protein expression profiles in response to environmental challenges. While this regulation is conventionally thought to occur primarily at transcription [11], several recent studies based on the translatome have revealed gene-specific translational regulation when bacteria are exposed to heat stress, oxidative stress, or amino-acid starvation [12–15]. These studies specifically focused on translational regulation in specific genes contributing to stress responses, such as the heat shock protein (HSP) gene family [16]. However, the universality of translational regulation as a response to general environmental challenges remains largely unexplored. For instance, nutrient limitations are fundamental challenges for microbes, and *E. coli* cells are known to adapt to different nutrient limitations by regulating ribosomal synthesis and resource allocation strategies [17–19]. But we do not yet know if cells also regulate the translation of specific genes in response to nutrient limitations. Recently, quantitative proteomic data provided valuable resources for the exploration of bacterial translational regulation under various environments [20]. However, clarifying the detailed process and mechanisms of translation regulation requires the integrated analysis of multiple omics, including transcriptomics, translatomics, and proteomics. Moreover, as transcription and translation are coupled in bacteria [21], it is worth quantifying the extent to which transcription and translation are regulated in concert to cope with environmental stresses.

To understand the general principles underlying the regulation of gene expression at the translational stage, quantitative models have been developed to provide an integrative picture of translational regulation [22–27]. However, the factors that contribute to gene-specific translational regulation upon environmental changes are still poorly understood [28]. In this regard, some researchers have suggested that protein synthesis rates are tightly linked to tRNA

composition and modification, which is known to vary across conditions [29–31]. However, others suggest that mRNA structure itself is a key factor in translational regulation, and contributes to rapid adaptation to changing environments by dynamically folding in response to multiple signals, including temperature, ligands, regulatory proteins, and small RNAs [32,33]. Indeed, translation efficiency is highly correlated with ORF mRNA structure rather than other mRNA features such as tRNA adaptation index (tAI) [34]. Therefore, the underlying mechanisms of translational regulation in response to environmental changes are worth exploring.

Here we extended our previous research [18] and systematically quantified the total number, bound fraction, and elongation rates of ribosomes under carbon (C), nitrogen (N), and phosphorus (P) limitations at different growth rates, and several other growth conditions. Then we extended our perspective on translational regulation from the global scale to the level of individual genes. We aimed to examine whether there is gene-specific translational regulation in *E. coli* responding to different nutrient limitations, and then explore possible mechanisms. Combining global ribosome profiling with RNA-seq, we quantified the correlation between mRNA level and translation efficiency both across genes and across conditions, as well as the variability of translation efficiencies across conditions. We uncovered a diverse range of gene-specific translational regulations concerted with transcriptional regulations in response to environmental deficiencies. Intriguingly, several translational regulation genes are themselves subject to translational regulation, suggesting possible feedbacks. Further analysis suggested that codon usage may play an important role in gene-specific translational regulation. Using a random forest model, we quantified the contribution of codon usage towards condition-dependent translational regulation. This analysis revealed that the cross-condition variability tends to be an inherent feature of individual genes, independent of particular conditions. These findings expand our understanding of translational regulation in response to environmental changes, and suggest novel strategies for effective translation in future synthetic biology.

## Results

### Cells adapt to different nutrient conditions through global translational regulations

To explore the translational regulation in *E. coli* under different environments, we utilized 12 different growth conditions (Table 1). In an effort to focus on distinct yet stable steady-state growth conditions, we grew *E. coli* in chemostats with dilution rates of 0.1 and 0.6 h⁻¹ under limitations for carbon (C), nitrogen (N), and phosphate (P). We also grew two mutant strains in chemostats, Δ*rplA* and Δ*leuB*, with the same dilution rates of 0.1 and 0.6 h⁻¹. *rplA* encodes a component of the 50S ribosome subunit [35,36] and *leuB* is involved in leucine biosynthesis [37]. These two mutant strains thus enabled us to probe how single-gene mutations that disrupt distinct aspects of the translation process affect the overall pattern of translation. In addition, wild type *E. coli* was also grown in batch culture using both glucose minimal media and defined rich MOPS media, with measured growth rates of 0.9 and 1.8 h⁻¹, respectively.

*E. coli* was grown under glucose (C, carbon), ammonia (N, nitrogen) and phosphate (P, phosphorus) limited conditions in chemostats at two different dilution rates of 0.1 and 0.6 h⁻¹

**Table 1. List of the 12 different conditions for *Escherichia coli* in our measurements.**

| Condition ID | 1 | 2 | 3 | 4 | 5 | 6 | 7 | 8 | 9 | 10 | 11 | 12 |
|---|---|---|---|---|---|---|---|---|---|---|---|---|
| Nutrient limitation | C-limited | | N-limited | | P-limited | | glucose minimal | defined rich MOPS | N-limited for Δ*rplA* | | Leu-limited for Δ*leuB* | |
| Growth rate (h⁻¹) | 0.1 | 0.6 | 0.1 | 0.6 | 0.1 | 0.6 | 0.9 | 1.8 | 0.1 | 0.6 | 0.1 | 0.6 |
| Culture environment | chemostat | | | | | | batch culture | | chemostat | | | |

(equal to growth rates). ΔrplA and ΔleuB mutant strains were grown under ammonia and leucine limitations, respectively. Three biological replicates were performed for all the 12 conditions.

As observed in a previous study [18], P-limited cells consistently exhibited lower RNA-to-protein ratios than C-limited or N-limited cells. Δ*rplA* cells exhibited a higher RNA-to-protein (R/P) ratio than other conditions at the growth rate of 0.6 h$^{-1}$, consistent with the significantly reduced activity of ribosomes for Δ*rplA* cells [38]. All other conditions are located on a single line of R/P ratio versus growth rate (Fig 1A), consistent with the well-established linear relationship between growth rate and RNA-to-protein ratio in previous studies [18,19,39]. The free ribosome pools decreased as growth rate increased across all the nutrient-limited conditions (Fig 1B). Our previous study suggested that *E. coli* differentially tune multiple ribosomal features, including ribosome total number, elongation rate, and active fraction, to achieve the same growth rate of 0.1 h$^{-1}$ under different nutrient limitations [18]. The current results confirmed that this pattern extends to a higher growth rate of 0.6 h$^{-1}$ (Fig 1C–1E). Meanwhile, under batch conditions, all three of these ribosomal features reached very high levels (Fig 1F). The distribution of ribosome density along mRNAs also revealed differences between conditions: For C- and N-limited conditions, there was a higher ribosome occupancy near the start codon, particularly at the lower growth rate (Fig 1G and the inset). Across all conditions, after the first few codons, the ribosome density exhibited no significant decrease along mRNAs (Fig 1G). In summary, cells adapt to different nutrient conditions by differentially tuning multiple ribosome-related features, which act globally on the translation efficiencies of all genes. In addition to such global translational regulation, we wondered whether there could be gene-specific translational regulation in response to different environment conditions.

## Quantifying transcriptome and translatome in *E. coli* under multiple nutrient limitations

To explore individual gene expression regulation in *E. coli* under various nutrient conditions, we quantified the genome-wide transcriptome and translatome by performing RNA-seq and global ribosome profiling for all the conditions above [40]. After filtering out ribosomal RNA (rRNA) and transfer RNA (tRNA) species, a total of 4321 genes were used as the reference for mapping.

Translation efficiency (TE) has conventionally been quantified as the rate of protein production per mRNA (the translating ribosome number per unit length of a mRNA) [41]. Here, a version of TE is obtained by dividing the rate of synthesis of each protein (ribosome density from ribosome profiling) by the mRNA levels from RNA-Seq under the corresponding condition; we call this quantity the "relative translation efficiency" (RTE, see Methods for details). It is worth noting that the RTE represents the relative occupancy of ribosomal resources devoted to translation, rather than the absolute protein production rate per mRNA molecule. To avoid the high noise caused by low mRNA levels, we filtered genes with a cut-off of $\log_{10}$(mRNA RPKM) > 1.5. After filtering, a total of 2914 genes were retained for further analysis. Scatter plots and correlation analysis of per gene mRNA and ribosome level showed high data reproducibility across different replicates (Fig A-B in S1 Text). Since RTE is the ratio of footprint densities to RNA-seq read densities, it could be sensitive to the changes of mRNA levels. To test whether RTE truly reflected differences in translation between genes, we analyzed the expression pattern of genes from the *dusB-fis* operon and the $F_0F_1$ ATP complex, which are two typical cases that controlled at translation level with similar mRNA abundances. *dusB* and *fis* are coregulated as part of the same operon, and we observed that their mRNA levels were comparable. However, because of the highly different mRNA structure [34], their RTEs

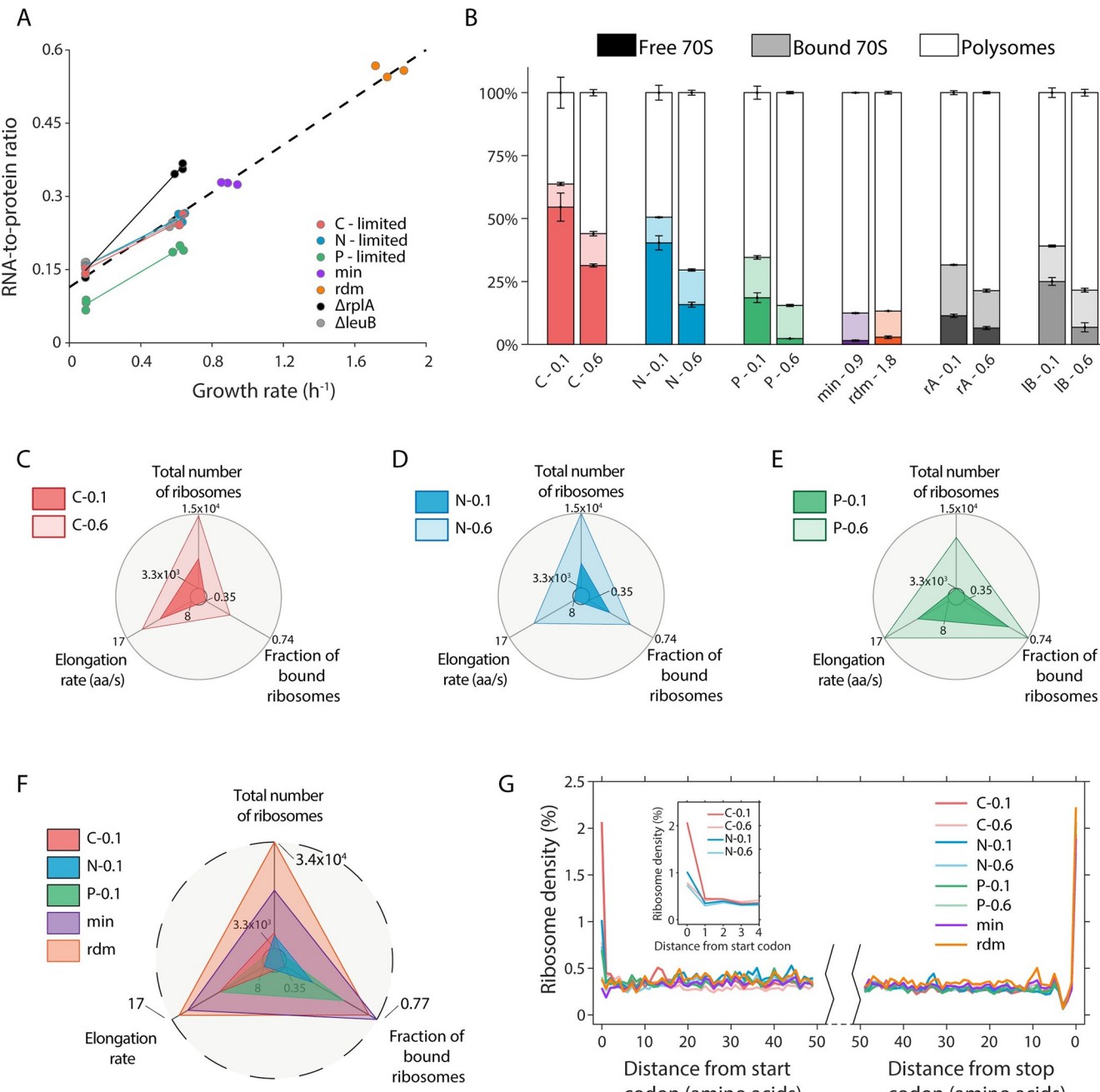

**Fig 1. Cells adapt to nutrient conditions through different ribosomal strategies.** (A) RNA-to-protein ratios for 12 conditions at different growth rates. Each data point represents one experimental measurement. (B) Fractions of assembled (70S) ribosomes under 12 conditions. The assembled ribosomes include free 70S monosomes, mRNA-bound 70S monosomes, and mRNA-bound 70S polysomes (multiple ribosomes on one mRNA). Free 70S, bound 70S, and polysomes are represented in dark, light, and white colors, respectively. The bar heights represent mean values with error bars indicating s.e.m. from three biological replicates. (C-E) Cells differentially regulate three ribosomal features in response to C-, N-, and P-limitations at the growth rates of 0.1 and 0.6 h$^{-1}$. The three features include total number of ribosomes per average cell (see Methods), elongation rate, and fraction of bound ribosomes. These features are scaled linearly between the inner circle and the outer circle, which represent the minimum and maximum among all conditions, respectively. The scales of the three indicated axes are the same for panels C-E. (F) Same as (C-E), but showing the differences between chemostat cultures and two batch conditions. The value of the outermost circle is larger than in C-E, especially the total number of ribosomes. (G) Averaged A-site ribosome density within the first and last 50 codons of the transcripts, from ribosome profiling analysis. Each curve shows the mean value from three replicates at each condition. Inset: ribosome density at the beginning of the transcripts.

showed significant disparities (Fig C in S1 Text), consistent with the results of previous study [34]. Eight subunits of the $F_0F_1$ ATP complex were from a single polycistronic transcript, and the mRNA levels of these genes were similar (Fig C in S1 Text). However, their RTEs varied substantially and were proportional to their stoichiometry in the $F_0F_1$ ATP complex (Fig C in S1 Text), consistent with a previous study by Li *et al.* [17] on the quantification of absolute translation efficiency. Thus, these results confirmed the reliability of RTE in quantifying translational differences among genes.

## mRNA-RTE correlation analysis suggests the preponderance of both gene-specific and condition-specific translational regulation

To quantify the correlation between mRNA level and RTE, we analyzed two types of correlation: across genes and across conditions (Fig 2A). For cross-gene correlations, we confirmed that, on average, mean mRNA levels positively correlated with RTEs (Fig 2B), with a coefficient of determination ($R^2$) of 0.3. This means that in an average sense, if one gene has a higher mRNA level than another, it is also likely to have a larger RTE.

Next, we wondered whether the mRNA levels and RTEs of individual genes change in concert across different conditions. To answer this question, we examined the cross-condition correlation between mRNA level and RTE for each gene. In contrast to the positive cross-gene correlation, we found a broad distribution of the 2914 Spearman's rank correlation coefficients between a gene's mRNA levels and its RTEs across the 12 conditions (Fig 2C, blue). The distribution ranged from -1 to +1, asymmetrically biased toward negative values. The median of this distribution was -0.23, and 24.5% of the genes exhibited a smaller than -0.5 correlation between their mRNA levels and RTEs across conditions; only 4.7% of the genes exhibited a larger than 0.5 correlation. To test the significance of this asymmetric and mostly negative distribution, we randomly scrambled the RTEs among conditions for each gene and recalculated the 2914 correlation coefficients to obtain a null distribution (Fig 2C, grey). As confirmed by theoretical analysis, this null distribution was symmetric with zero mean (see S1 Text for details), and visibly distinct from the actual distribution. The sizable, statistically significant difference between the actual distribution and the null distribution implied the widespread existence of gene-specific translational regulation, where the RTE of an individual gene changed in response to different environmental conditions (Fig D in S1 Text).

To further explore the possible roles of gene-specific translational regulation, we examined two genes with highly negative and highly positive mRNA-RTE correlations. For the gene *fieF*, which mediates metal-ion transport in response to iron poisoning [42], the correlation coefficient was -0.91 (Fig 2D). Iron homeostasis is essential for cell survival [42]. For the gene *ycaO*, which is involved in the β-methylthiolation of the ribosome complex S12 [43], the correlation coefficient was 0.73 (Fig 2E). Ribosome abundance has been known to change with growth conditions or cellular status [44]. In addition, the top 5 genes with negative or positive correlations between mRNA level and RTE are shown in Fig E in S1 Text. Among them, *cutC* and *fieF* with negative correlation are involved in the maintenance of homeostasis, while *phnG* with positive correlation is involved in the utilization of phosphorus. These examples raise a more general question: Do genes with negative and positive mRNA-RTE correlations perform different biological functions?

## Correlations between mRNA level and RTE link to gene function

To test the hypothesis that genes with distinctive mRNA-RTE correlations fall into different functional categories, we performed gene ontology (GO) enrichment analysis for the top 300 genes with the strongest negative mRNA-RTE correlations across conditions, as well as the top

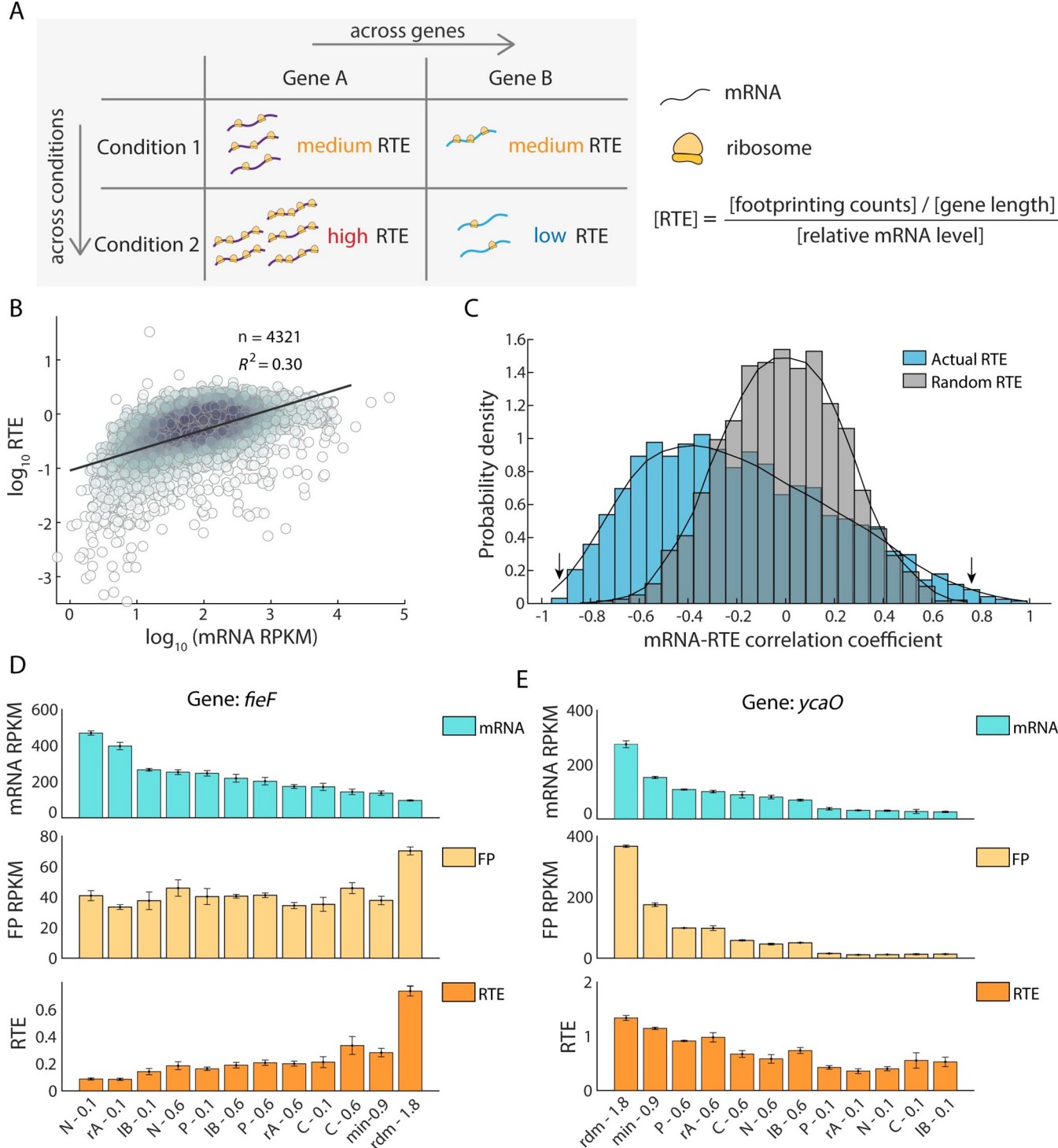

**Fig 2. Global view of mRNA-Relative Translation Efficiency (RTE) correlations across genes and across conditions.** (A) Two types of correlations between mRNA level and RTE: across genes and across conditions. (B) Correlation between mean mRNA level and mean RTE, across different genes. Mean levels were taken as the average of all 12 conditions (Table 1). Each dot represents one gene, and color depth depicts the density of points. (C) Distribution of Spearman's rank correlation coefficients between mRNA level and RTE across the 12 different conditions. Each gene provides one such correlation coefficient, and distributions are shown for 2914 genes (blue bars–Actual RTE: original RTE from RNA-Seq and Ribosome profiling; gray bars–Random RTE: the RTE values for each gene were randomly scrambled among the 12 conditions to obtain the randomly ordered RTEs, which was considered as the null distribution, see methods and S1 Text for details). The $p$-value between the two distributions from Kolmogorov-Smirnov test was 5e-130. (D-E) Example of two genes with negative correlation (D, gene *fieF*, left arrow in B) and positive correlation (E, gene *ycaO*, right arrow in B) between their mRNA levels and RTEs. The mRNA

level (up panel), ribosome footprints (middle panel), and RTE (down panel) under 12 conditions were shown. Error bars represent s.e.m. from three biological replicates.

300 genes with the strongest positive correlations. We found that the top 300 most negatively correlated genes were mainly involved in biological processes that may not be affected by our nutrient limitations (Fig 3A). These genes, such as *nagC*, *ascG*, and *kdgR*, are essential for the homeostasis of metabolism. Other genes in this group, such as *rpoE*, *lacI*, and *frmR*, respond to inputs such as heat shock, lactose, or formaldehyde which were not assayed in our experimental condition. It is conceivable that the negative correlation between mRNA level and RTE for genes in this group reduces the dependence of protein abundance on conditions.

By contrast, the top 300 genes with the strongest positive correlation were mainly involved in nutrient utilization, stimulus response, and translational regulation itself (Fig 3B). These genes are the key cellular factors that respond to the imposed nutrient limitations. Interestingly, the observation of strong positive correlation for genes involved in translational regulation hinted at possible direct feedback, i.e. that genes regulating translation are themselves subject to translational regulation. Take two typical genes as examples: *rplA*, encoding a component of the 50S ribosome subunit, functions in translational regulation [35,36]. The mRNA level of r*plA* was significantly upregulated at a growth rate of 0.6 h$^{-1}$ comparing to 0.1 h$^{-1}$. In concert, the RTE of *rplA* was also significantly upregulated at the faster growth rate (Fig 3C). This phenomenon was robust under all three nutrient limitations, C, N, and P. Similarly, *rmf*, a translation inhibitor, is also subject to translational regulation. RMF is a ribosome modulation factor that reversibly converts active 70S ribosomes to a dimeric form, which is associated with a decrease in overall translational activity during the transition from exponential growth to stationary phase [45]. Our data show that for *rmf*, both mRNA level as well as RTE were significantly down-regulated at faster growth rates, regardless of which nutrient was limiting (Fig 3D).

## Gene-specific translational regulation in response to nutrient limitations

Despite the mostly negative distribution of mRNA-RTE correlations for all genes, our former analysis suggests concerted regulation of both mRNA level and RTE for genes responsive to environmental changes. To systematically examine such concerted regulation, we analyzed the relative changes at both mRNA and RTE levels between pairs of nutrient limitations under the same growth rate of 0.1 h$^{-1}$. We first compared the expression between C-limited and N-limited cells. We used a cutoff of $\log_2$(C-/N-limited mRNA fold change) $> 4$ and *p*-value $< 0.05$ to select a group of differentially expressed genes at the mRNA level. For genes with significantly upregulated mRNA levels under C limitation, 83.3% (40/48) of their RTE fold changes were also greater than 1, thus exhibiting concerted regulation of transcription and translation (Fig 4A, red dots). In the same way, genes with significantly upregulated mRNA levels under N limitation also showed upregulated RTEs (Fig 4A, blue dots). Similar phenomena can be observed when comparing N-limited and P-limited cells: for genes with upregulated mRNA levels under P limitation, their RTEs were also significantly upregulated (Fig 4B, green dots), and similarly for genes upregulated under N limitation (Fig 4B, blue dots–same genes as in Fig 4A). Next, we compared the mRNA level and RTE across C, N, and P limitations in parallel for the three gene groups selected above. The results further confirmed that mRNA and RTE change in concert for genes that are specifically expressed under specific nutrient limitations (Fig 4C–4E).

To verify whether these three groups of genes are actually involved in utilization of specific nutrients, we performed GO analysis for each group. The resulting functional enrichment

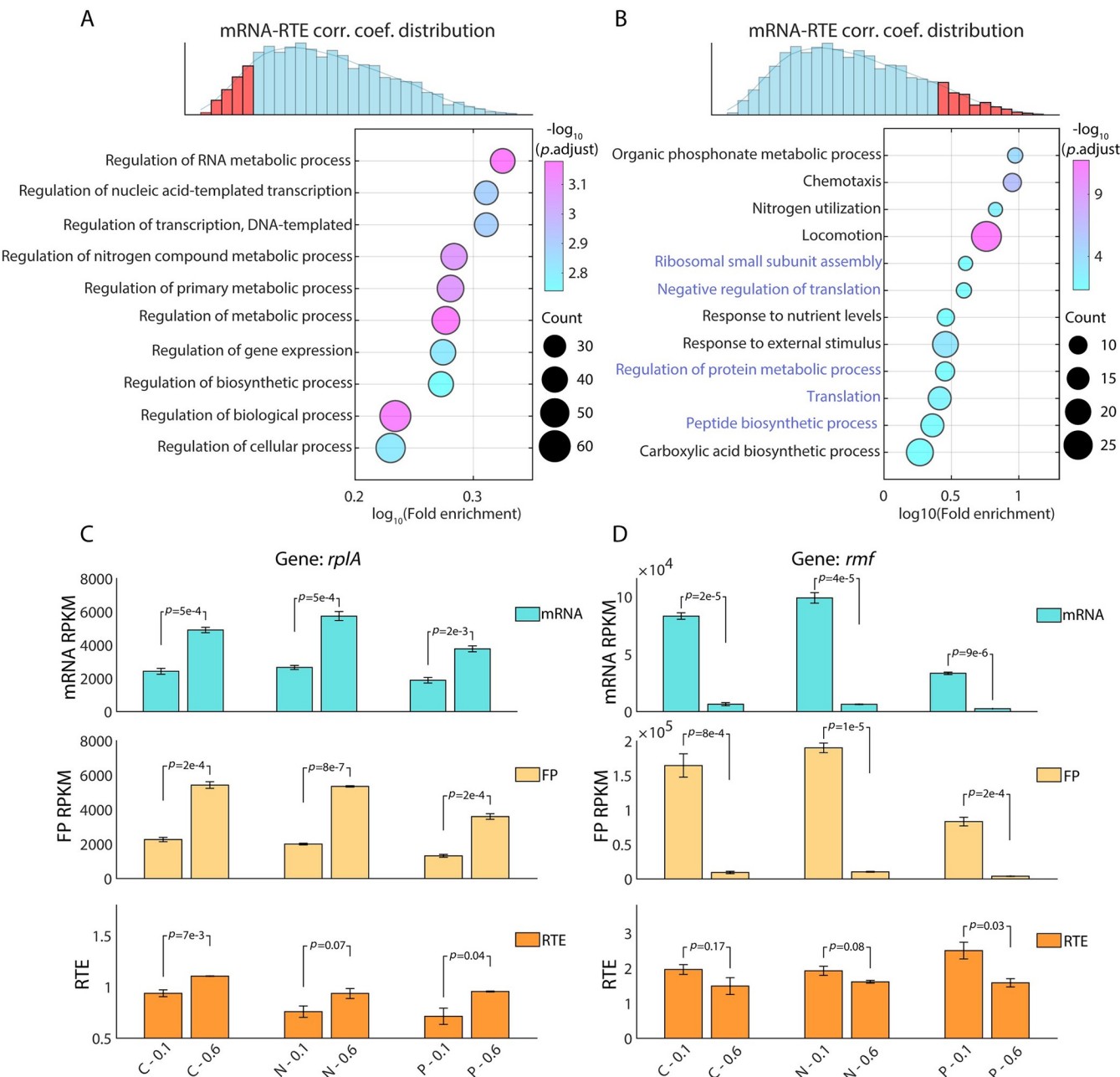

**Fig 3. The correlation between mRNA level and RTE is related to specific gene functions.** (A-B) Gene ontology (GO) enrichment for the top 300 genes with the most negative correlation (A) and the most positive correlation (B) between mRNA levels and RTEs. The color of the dots represents the -$\log_{10}$ adjusted $p$-value, and the dot size represents the number of genes appearing in each biological process. (C-D) Genes regulating translation are themselves subject to translational regulation. Examples of positive correlation between mRNA level and RTE across different growth rates for one gene that promotes translation (C, gene *rplA*) and one that inhibits translation (D, gene *rmf*). The mRNA level (up panel), ribosome footprints (middle panel), and RTE (down panel) under 12 conditions were shown. Student's *t*-test was used to calculate the $p$-value. Reads Per Kilobase Million (RPKM) is used for mRNA level and ribosome footprints.

confirmed our hypothesis. Genes with concerted upregulation of both mRNA level and RTE under C limitation were mainly involved in the transport of carbon-containing compounds or cell locomotion (Fig 4F). For example, the gene *yjcH* (marked in red in the upper right region of Fig 4A) encodes a protein involved in acetate catabolism and transport [46], while the genes

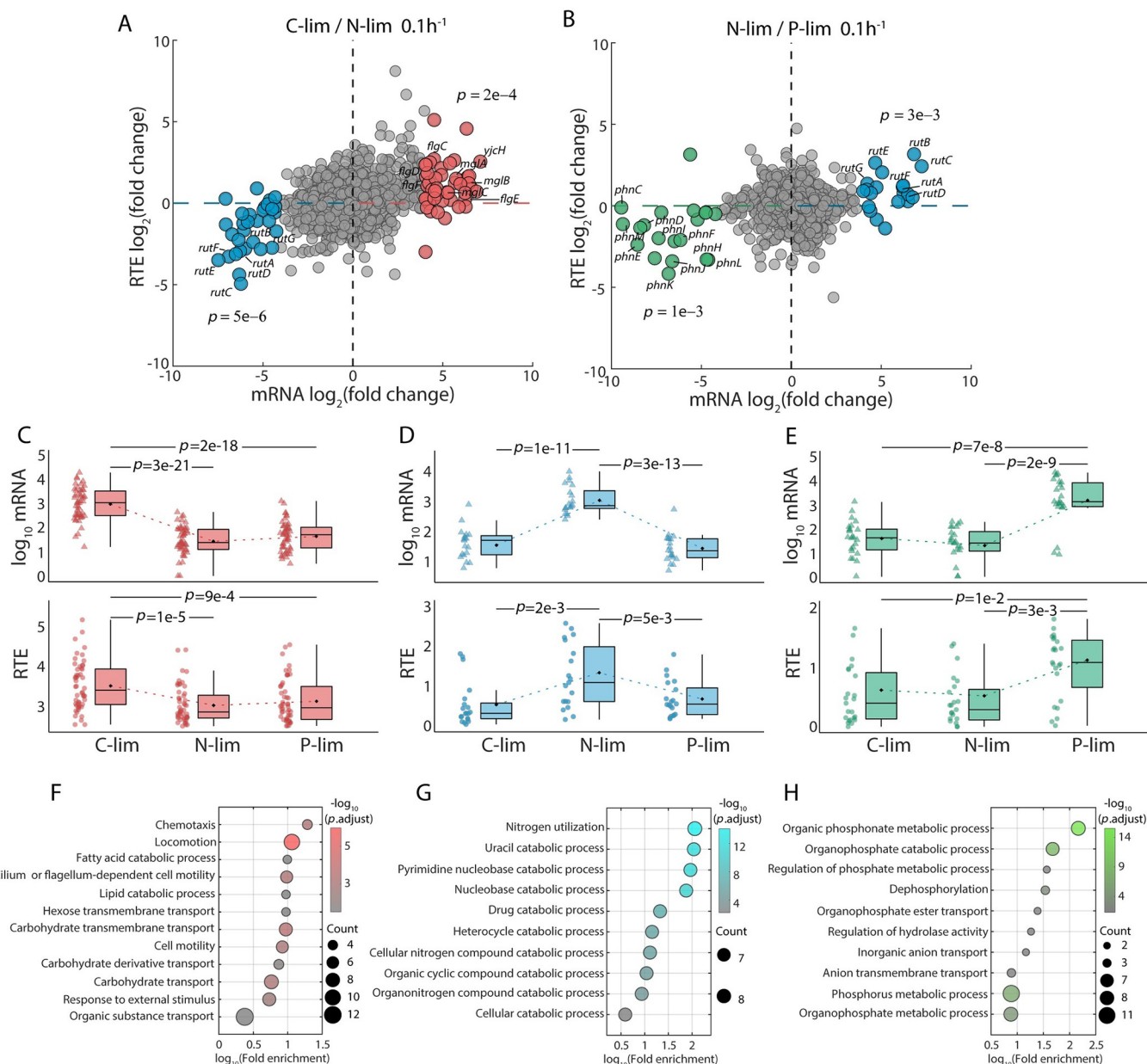

**Fig 4. Transcription and translation couple together to respond to nutrient limitations.** (A) Comparison of transcription changes (log$_2$ mRNA fold change, *x*-axis) and translation changes (log$_2$ RTE fold change, *y*-axis) between carbon limitation and nitrogen limitation at the growth rate of 0.1 h$^{-1}$. The averages of three biological replicates are shown. Red dots represent genes with log$_2$ mRNA fold change (C-limited / N-limited) > 4. Blue dots represent genes with log$_2$ mRNA fold change (C-limited / N-limited) < -4. *p*-value was used to test the significance of the RTE fold change between the highlighted genes and the background genes. (B) Same as (A), but showing the change between nitrogen limitation and phosphate limitation. Green dots represent genes with log$_2$ mRNA fold change (N-limited / P-limited) < -4. (C-E) mRNA level (upper panel) and RTE level (lower panel) of the three groups of highlighted genes in (A) and (B). The three highlighted groups of genes are upregulated under carbon (C), nitrogen (D), and phosphate (E) limitations. (F-H) Gene ontology (GO) enrichment analysis for the three highlighted gene groups in (A) and (B). The color of the dots represents the -log$_{10}$ adjusted *p*-value, and the size represents the number of genes.

*mglA-C* are involved in galactose transport, which also responds to C limitation [47]. Other genes in this group, *flgC-F*, are involved in flagellar assembly. Genes with concerted upregulation of both mRNA level and RTE under N limitation were mainly involved in nitrogen utilization (Fig 4G). The genes *rutA-G* (marked in blue in the lower-left region of Fig 4A) in the

*rut* pathway are typical examples: they contribute to derivation of nitrogen from pyrimidines [48, 49]. Similarly, genes with concerted upregulation of both mRNA level and RTE under P limitation were mainly involved in phosphorus metabolism (Fig 4H). These genes mainly consist of the *phn* gene cluster (marked in green in the lower-left region of Fig 4B), which is induced under phosphate limitation and plays an important role in deriving phosphate from phosphonate degradation [50,51]. The same analyses were performed for the growth rate of 0.6 h$^{-1}$, with consistent results, except for no significant observed difference in RTE between C and N limitations for the genes involved in nitrogen utilization (Fig F in S1 Text).

To test for gene-specific translational regulation between different growth rates, we performed similar analyses at the growth rates of 0.1 h$^{-1}$ and 0.6 h$^{-1}$ under the same nutrient limitation. Intriguingly, there were no evident differences in translational regulation between the two different growth rates (Fig G in S1 Text). Even the highlighted subsets of genes identified in Fig 4, involved in utilization of specific nutrients, showed no significant difference in RTE between different growth rates under the same types of nutrient limitation (Fig G in S1 Text). In summary, these results strongly suggest gene-specific translational regulation in response to different nutrient limitations but not different growth rates.

## Gene functions associate with translational regulation patterns

Our findings revealed a diverse spectrum of translational regulation in response to different nutrient limitations but not in response to different growth rates. Next, we wanted to explore whether different translational regulation patterns were associated with certain genetic functions. The genes involved in the assimilations of C, N, and P all exhibited specific upregulation in RTE under specific nutrient limitations, implying the existence of one class of genes–those whose RTEs changed across conditions and, as a result, exhibited high RTE variability. Meanwhile, there could also be another class of genes with stable RTEs and consequently low RTE variation. Therefore, we wanted to investigate the possible sources of RTE variance (Fig 5A). To obtain a global view, we first calculated the mean and variance of RTE for each gene across the 12 different conditions (Fig 5B). Overall, the results exhibited an overall positive correlation between the mean and variance of RTE. However, genes with similar mean RTE still exhibited remarkable differences in their RTE variance.

To clarify the relationship between RTE patterns and function, we zoomed in from a function-related perspective. We compared the translational regulation patterns of the 82 pathways in *E. coli* [52], and found that several of them are distinguishable in the mean-variance biplot of RTE (Fig H in S1 Text). For example, four functional pathways occupied two distinguishable regions in the biplot (Fig 5B, colored dots). Compared with the overall transcription and translation pattern of background genes (Fig 5C), the TCA cycle and the pyruvate metabolism pathways shared similar translation patterns, with a small RTE variance (Fig 5D). These two pathways are both involved in basic metabolic processes [53,54]. By contrast, the flagellar assembly pathway and the bacterial chemotaxis pathway both exhibited large RTE variance (Fig 5E). In addition, their mean RTEs were significantly positively correlated across the 12 conditions (Fig 5F). These two pathways are both involved in cell motility [55,56].

## Patterns of relative translation efficiency associate with codon usage

The fact that functionally relevant pathways share similar RTE patterns inspired us to search for commonality between the above pairs of pathways with comparable functions and similar RTE patterns. Intriguingly, we found that the codon frequencies of genes in pyruvate metabolism and TCA cycle pathways were highly similar, with a Spearman's rank correlation coefficient of 0.93 (Fig 6A). Also, genes in flagellar assembly and bacterial chemotaxis pathways

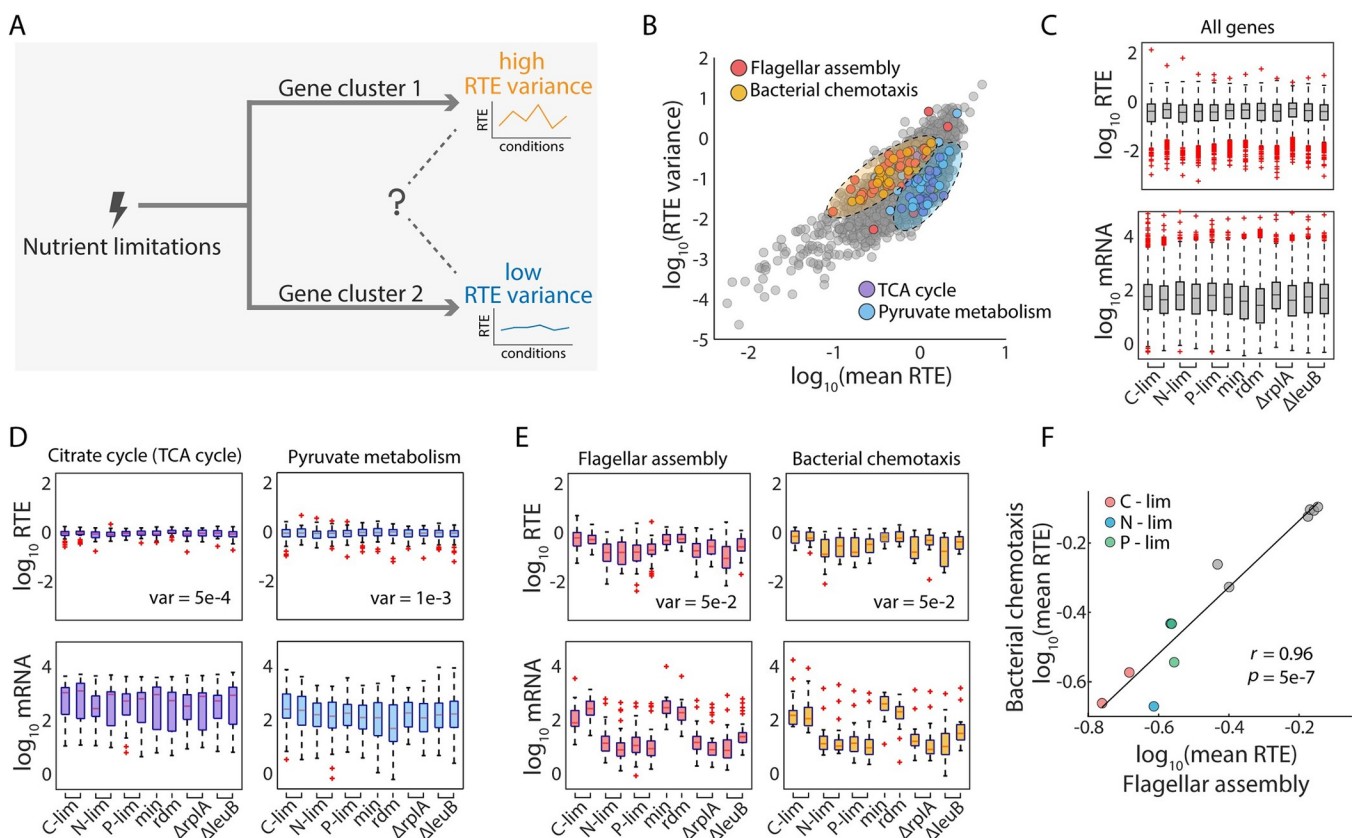

**Fig 5. Pathways with similar gene expression patterns share similar gene functions.** (A) Illustration of the question we investigated: when genes are classified by their cross-condition variance of RTE, what is the cause of this high-or-low variance classification? (B) Relationship between mean and variance of RTE across 12 conditions. Each dot represents one gene. Colored dots are genes involved in four selected pathways with different patterns of translational regulation. (C) Distribution of RTEs (upper) and mRNA levels (lower) of all genes, under the 12 different conditions. (D) Distribution of RTEs (upper) and mRNA levels (lower) for genes involved in TCA cycle (left panel) and those involved in pyruvate metabolism (right panel). (E) Distribution of RTEs (upper) and mRNA levels (lower) for genes involved in flagellar assembly (left panel) and those involved in chemotaxis (right panel). (F) Correlation of mean RTE across 12 conditions between genes involved in flagellar assembly and bacterial chemotaxis.

share similar codon usage and the Spearman's rank correlation coefficient between their codon frequencies was 0.52 (Fig 6B). As a control analysis, we randomly selected gene sets containing 50 genes (comparable to the number of genes in a typical pathway) and found that most of the correlation coefficients were in the range of -0.5~0.5, suggesting that the high correlation coefficients between the aforementioned pathways were meaningful (Fig I in S1 Text). Overall, most of the 82 KEGG pathways in *E. coli* had small RTE variance. Therefore, the fact that a few pathways exhibit large RTE variability under nutrient limitations may have important biological implications, notably for flagellar assembly, bacterial chemotaxis, and some aspects of biosynthesis and metabolism (Fig I in S1 Text). Interestingly, among the top 10 pathways with the highest RTE variance, codon frequency correlations are generally strong (Fig I in S1 Text), suggesting possible shared sets of codon patterns that enable genes to exhibit high variability in RTEs in response to environmental stresses.

The observation concerning specific pathways inspired us to quantify how much codon usage contributes to this cross-condition RTE variance. Overall, the mean codon frequencies for the top 200 genes with the largest RTE variance and those for the bottom 200 genes with the smallest RTE variance exhibited a negative correlation (Fig 6C, Spearman's rank correlation coefficient -0.55). Globally, certain codons appeared with changing frequencies for genes

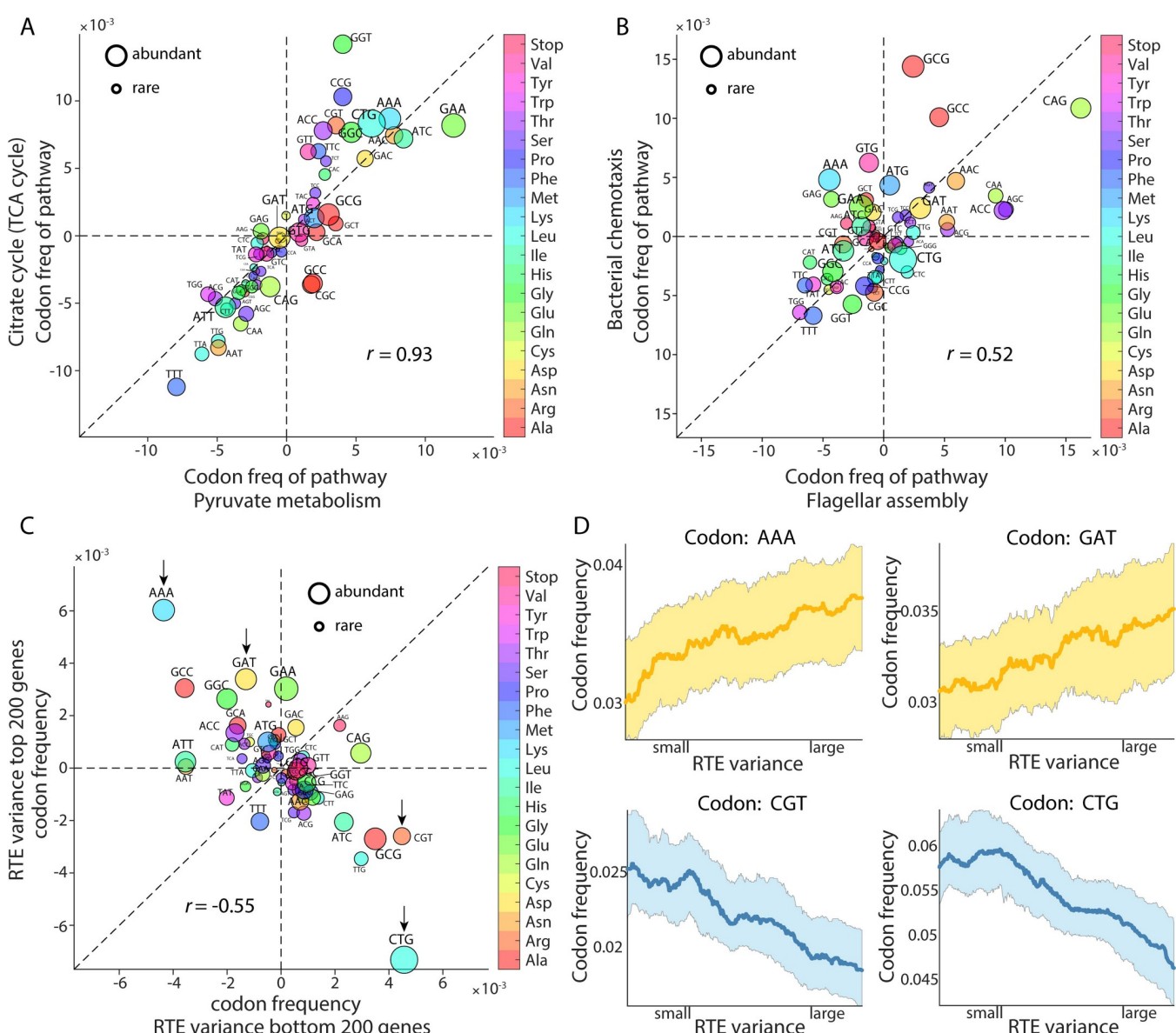

**Fig 6. The relationship between codon usage and gene RTE patterns.** (A) Correlation of codon frequencies between genes in the two pathways described in (Fig 5B). After removing the overlapped genes, there are 38 and 13 genes involved in each pathway, respectively. The 64 codons are dotted with sizes representing the rarity of codons in *E. coli*. (B) Correlation of codon frequencies between genes in the two pathways described in (Fig 5C). After removing the overlapped genes, there are 38 and 15 genes involved in them, respectively. (C) Negative correlation between the codon frequencies for the top 200 and bottom 200 genes in their RTE cross-condition variances. Four anti-correlated codons are indicated by arrows. (D) Relationship between the gene-by-gene RTE variance (*x*-axis) and codon frequencies (*y*-axis), for the four codons highlighted in (C). The average over 2914 genes is shown. The shadings represent the fluctuation of codon usage frequencies and the highlighted lines show smoothed mean results.

with different RTE variabilities. We singled out four codons with discrepant frequencies between the high-RTE-variability genes and the low-RTE-variability genes: The frequencies of AAA and GAT in the 2914 genes showed an overall increasing trend with increasing RTE variance (Fig 6D, upper). On the other hand, the frequencies of CGT and CTG showed an overall decreasing trend with increasing RTE variance (Fig 6D, lower). Interestingly, none of them is a rare codon. Actually, the nine rarest codons in *E. coli* (AGA, AGG, ATA, CCC, CGA, CGG, CTA, GGA, and TTA) showed no obvious bias in the two sets of genes with largest and

smallest RTE variance [57]. By contrast, while the mean codon frequencies for the top 200 genes with the highest mRNA levels and those for the bottom 200 genes with the lowest mRNA levels also exhibited negative correlation (Fig J in S1 Text, Spearman's rank correlation coefficient -0.30), the nine rarest codons were enriched in genes with low mRNA levels (Fig J in S1 Text). This suggests that the RTE changes mediated by codon usage in our study are distinct from the low translation efficiency mediated by rare codons.

## Codon usage partially predicts the cross-condition variability of RTE

Beside codon usage, RTE variance positively correlated with the mean value of RTE. In our data, there was also a weak correlation between RTE variance and mRNA level. Therefore, we needed to carefully separate the influences of the absolute value of RTE and mRNA level to examine whether codon usage directly contributes to the cross-condition RTE variability. We utilized a random forest model to quantify the contribution of different features to the prediction of RTE variance. The flowchart of the algorithm is shown in Fig 7A. First of all, according to the median of RTE variance, we divided the 2914 genes into two clusters, which represent large and small RTE variance respectively. Then 80% of the genes were randomly sampled as the training set, leaving 20% as the test set. For the training set, Breiman's random forest algorithm was used to train a random forest model until the error converged. Different combinations of the features were separately used for training. By comparing the results from different feature combinations used for classification, we were able to quantify how much each single feature contributes to RTE variance. The receiver operating characteristic (ROC) curves suggested that the absolute value of RTE contributes most of the classification accuracy (Fig 7B, yellow line). The addition of the feature mRNA level only improved the classification accuracy slightly (Fig 7B, purple line). Nevertheless, the addition of the feature codon frequency improved the classification accuracy by approximately 10% (Fig 7B, red line, and Table 2), suggesting a nonnegligible and independent contribution from codon frequency to the cross-condition RTE variability.

An advantage of random forest models is that the contribution of each feature to the classification result can be quantified. The rank of codons contributing to classification from our random forest model (Fig K in S1 Text) is consistent with the anti-correlated codons in Fig 6C.

Furthermore, we examined whether other features contribute to RTE variance, such as the distribution of the third base for codons, gene length, and translation pause motifs consisting of adjacent double or triple codons [58–60]. The results showed that these features have little effect on classification accuracy (Fig K and Table A in S1 Text). In addition, we used two other evaluation indices to test whether the conclusion was robust with respect to different definitions of RTE variability: the Fano factor and the coefficient of variation (CV, see Methods). In both cases, the addition of the feature codon frequency markedly improved the classification accuracy (Fig L in S1 Text), consistent with our results using the index of RTE variance. In summary, codon usage contributed to the cross-condition RTE variability of genes and the result was robust according to our tests.

## Codon-related RTE variability is an inherent feature of genes

An intuitive hypothesis is that codon-related RTE variability could be due to the adaptation of tRNA pools to the environment. Indeed, codon usage has been suggested as a mechanism of translational regulation under oxidative stress or heat shock, as codon usage can be coupled to environment-dependent factors such as the tRNA pool composition [30,61]. An analogous extrapolation to explain our observed codon-related RTE variability would be as follows:

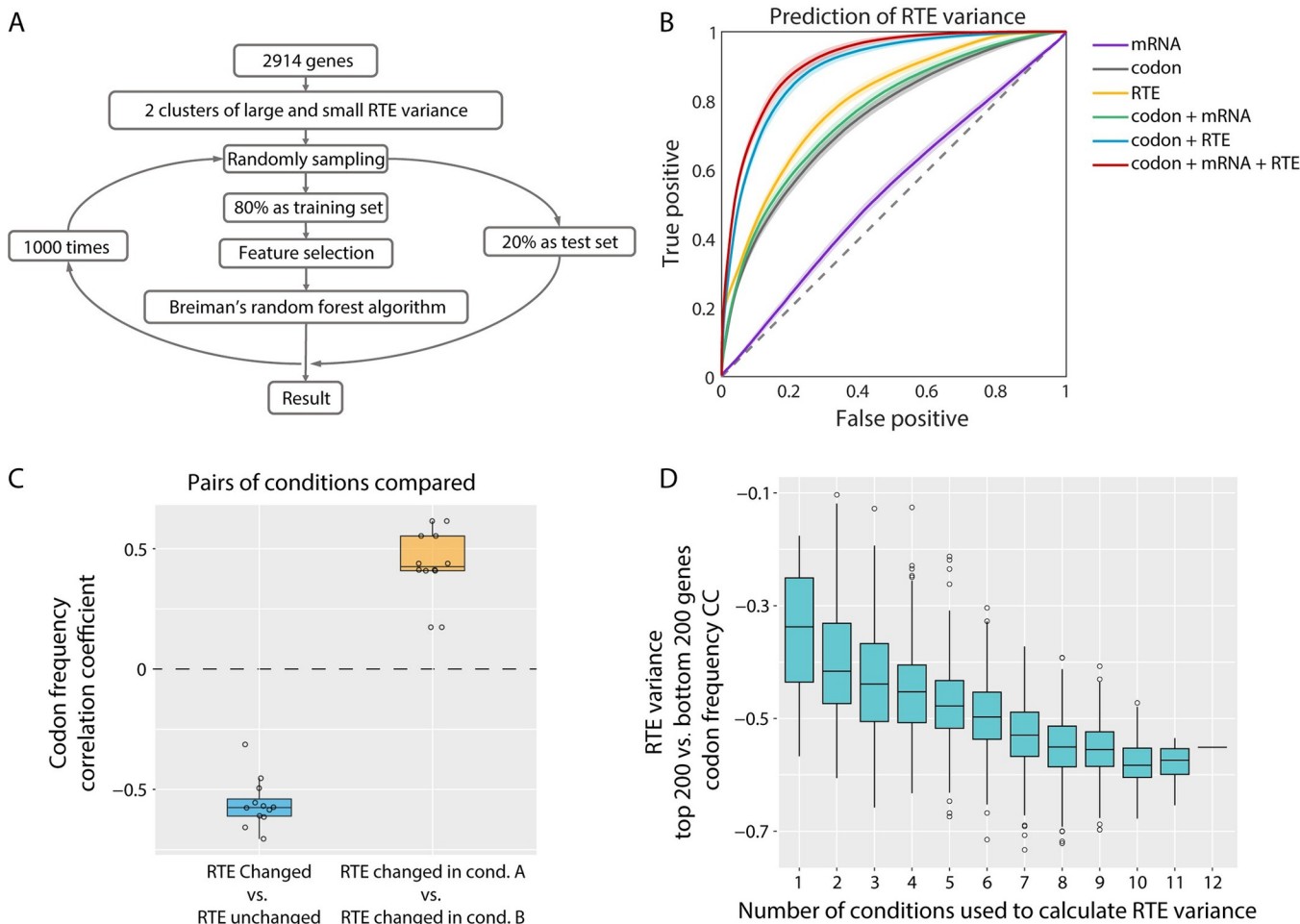

**Fig 7. Codon usage contributes to RTE variability across nutrient conditions.** (A) Flowchart for predicting the classification of RTE cross-condition variance using a random forest model. (B) The ROC curves of the classification accuracy using different combinations of features. An average of results for 1000 trainings was used. The shaded areas represent the S.D. Codon, mRNA, and RTE stand for the codon frequency, the mRNA level, and the RTE absolute value, respectively. (C) Codon frequency correlations (evaluated as in Fig 6C) between different gene sets when pairs of conditions were compared (Fig M in S1 Text). The yellow box shows correlations between two sets of RTE up-regulated genes in each of the paired conditions. The blue box shows correlations between RTE up-regulated and unchanged genes in each of the paired conditions. (D) The correlation coefficient was exactly as obtained in (Fig 6C), but for different numbers of conditions used to calculate the RTE variance. Each box was derived from all possibilities of taking n from the 12 conditions.

**Table 2. Classification results of the random forest model.**

| Features | Sensitivity | Specificity | Accuracy | AUC |
|---|---|---|---|---|
| mRNA | 53.4±3.19% | 53.47±3.08% | 53.41±1.81% | 0.54±0.02 |
| Codon | 64.27±3.08% | 71.96±2.9% | 68.07±1.76% | 0.75±0.02 |
| RTE | 72.58±2.68% | 72.5±2.61% | 72.52±1.61% | 0.8±0.02 |
| Codon + mRNA | 66.44±3.07% | 72.53±2.91% | 69.45±1.87% | 0.77±0.02 |
| Codon + RTE | 81.95±2.33% | 81.53±2.31% | 81.72±1.52% | 0.89±0.01 |
| Codon + mRNA + RTE | **84.36±2.19%** | **82.99±2.24%** | **83.66±1.5%** | **0.91±0.01** |

The table shows the average results with S.D. from a thousand random samples.

different nutrient conditions lead to distinct compositions of the tRNA pool, so that genes with codon frequencies matching a particular tRNA pool would have increased translation efficiency in the corresponding nutrient condition, thus producing high cross-condition variability. This hypothesis predicts that codon-related RTE variability would be condition-dependent. That is, there would be different sets of codons for high RTE genes for each nutrient condition, and the identification of "high variability genes" would depend on which conditions are being compared.

However, this hypothesis was found to test negative in our dataset. We compared the C-, N-, and P-limited conditions in pairs. When any pair of conditions A and B were compared, genes with a significantly higher RTE in A and those with significantly higher RTE in B actually share similar codon frequencies: there are no "condition-specific" codons that distinguish high-RTE genes in A from those in B (Fig 7C, M in S1 Text). By contrast, negative correlations of codon frequencies were observed between highly variable RTE genes and stable genes, between any pairs of conditions (Fig 7D). These observations indicate that genes can be divided into two classes according to their RTE variability, which has to do with their codon usage, but is independent of nutrient conditions.

To further confirm that the codon-related RTE variability does not rely on specific conditions, we randomly selected sets of conditions from the 12 conditions to calculate RTE variance. Then the top 200 and bottom 200 genes of RTE variance were used to calculate the correlation coefficient of codon frequency. We found a clear downward trend of the correlation coefficient with increasing number of conditions, asymptoting to a strongly negative correlation of $r \sim -0.55$ when more than 8 conditions were picked. This indicates that codon-related RTE variability is an inherent feature of genes that applies across multiple conditions.

## Discussion

How well transcript level represents protein abundance remains a controversial issue [2,3]. Translational regulation is one of the key factors affecting the correlation between transcript level and protein abundance in bacteria [2]. In this work, we systematically examined the ribosomal behaviors in response to various nutrient conditions. Then combining ribosome profiling and RNA-seq in *E. coli*, we quantified genome-wide RTE under 12 conditions and observed a diverse range of gene-specific translational regulations in response to nutrient conditions. Furthermore, using a random forest model, we discovered that codon usage partially predicts the cross-condition RTE variability, such that a particular subset of codons, especially AAA (Lysine) and GAT (Aspartate), favors variability across all the nutrient conditions. By contrast, CGT (Arginine) and CTG (Leucine) disfavor RTE cross-condition variability (Fig K in S1 Text). These findings broaden the understanding of translational regulation under environmental changes. What is more, our quantification of the contribution of codon usage to translational regulation can assist in the design of effective translation strategies in synthetic biology, as well as guide theoretical efforts to predict gene expression in response to environmental changes.

One important note is that the notion of RTE used in this work is slightly different from the TE in previous studies [17,41]. RTE represents the relative ribosomal resources allocated by per unit length of mRNA molecules. It does not stand for the absolute translation efficiency (TE), which also includes global translation-related factors such as the total number, the working fraction, and the elongation speed of ribosomes. These global factors affect the TE of all genes as a whole [18], while RTE involves translational differences between individual genes. Therefore, by quantifying RTE, we capture the ribosomal resources devoted to translation at the single-gene level, and thus can compare translational regulation among different

conditions, excluding the effect of global translation-related factors. In fact, according to comparison with previous studies on the translation efficiency of operons [17,34], RTE reliably reflects translation differences between genes (Fig C in S1 Text).

Protein biosynthesis consumes a large amount of building blocks and energy in fast-growing bacteria [62]. To ensure efficient allocation of translation resources and so maximize cell growth, the protein synthesis rate is precisely controlled in proportion to the stoichiometry of complexes or hierarchical functions [17]. We found that the overall mRNA-RTE correlation across genes is not affected by mutations in single genes such as Δ*rplA* and Δ*leuB* which are involved in translation processes (Fig N in S1 Text). This positive mRNA-RTE correlation reflects inherent translational differences along with the transcriptional differences between genes. By contrast, the mRNA-RTE correlation of a single gene under multiple conditions reflects whether the mRNA and RTE of the gene change consistently when the environment changes. Previous studies reported gene-specific translational regulation in bacteria under various stimuli [12,13], which enables a faster response to environmental stresses than through transcriptional regulation [2,63]. In our findings, both negative and positive mRNA-RTE correlations are likely biologically meaningful. For genes not sensitive to environmental changes, the mRNA level and RTE may be negatively correlated to stabilize protein production rate. For genes responding to specific nutrient limitations, the RTE may positively correlate with its mRNA level to amplify the change of protein synthesis rate, thus leading to a stronger correlation between mRNA level and protein abundance [64]. Gene-specific translational regulation is observed under C-, N-, and P-limitations. Therefore, the concerted regulation of transcription and translation may be a general strategy for cells to amplify their adaptation to environmental changes. In addition, the variance of RTE across conditions displays a large range, indicating that different genes are subject to varying degrees of translational regulation.

Also, according to our data, we suspected that translational regulation not only acts on genes responding to specific stressful conditions, but also acts on genes regulating translation itself, forming possible feedbacks [65]. Studies have revealed certain ribosomal proteins as feedback regulators, such as L1, S4, and S7 [66,67]. Previously, this kind of feedback regulation was believed to be associated with growth-rate-dependent ribosome synthesis [35]. In our findings, the RTEs of several proteins involved in translational regulation correlate strongly with their mRNA levels, indicating concerted translational regulation. Feedback regulation on translation allows for better regulation in the overall translation activity of cells, providing one additional possible strategy for bacteria to rapidly and effectively respond to environment changes.

Studies have suggested that rare codons do not limit translation efficiency in nutrient-rich media [68]. By contrast, under stresses such as nutrient limitation and oxidative stress, translation efficiency becomes sensitive to codon-usage-mediated tRNA dynamics [69, 70]. However, the details of how translation efficiency changes under stress remain unclear [71]. Our analysis suggested that codon usage not only contributes to condition-independent translation efficiency, but also partially predicts the variability of RTE across conditions. For condition-independent translation efficiency, multiple factors encoded in mRNA sequences affect the initiation, elongation, and termination of translation [5–7]. In particular, genome-scale studies have revealed significant association between codon usage and translation efficiency [72]. Codon usage *per se* mainly contributes during the elongation process [73], as it couples translation rates to the composition of the tRNA pool. However, our analysis indicated that under environmental stresses, the codon-related RTE variability across conditions was an inherent feature of genes, independent of specific conditions. Therefore, such RTE variability cannot be simply attributed to coupling between codon usage and the tRNA pool under any specific nutrient condition. This finding is consistent with the speculation of a previous study that the

change in tRNA composition leads to different translation efficiencies between stress-response and non-stress-response genes [30].

One limitation of our study is the lack of a detailed mechanism for how codons contribute to gene-specific translational regulation. As the translation process from mRNA to protein involves many factors, the differences in codon frequencies among mRNAs cannot be directly mapped to differences in translation efficiencies. In fact, it has been reported that there are complex interactions among multiple factors affecting translation, making it difficult to characterize the relation between codon frequency and translation efficiency [28]. For example, trade-offs between tRNA-mediated codon selection and mRNA structure entangle their separate roles [74]. Therefore, it remains an intriguing puzzle how codon frequency, a condition-invariant innate property of a gene, influences a gene's ability to respond to different conditions. We believe that in future research, a combination of technical approaches such as tRNA sequencing, mRNA structure probing, and translation-site-specific ribosome profiling will help uncover more mechanistic features of translational regulation [75,76].

## Methods

### Cell strains and growth conditions

Escherichia coli strain NCM3722 was grown in batch or continuous cultures. Dilution rates of 0.1 h-1 and 0.6 h-1 were used to define slow and fast growth rates in chemostats. We utilized a 300mL volume chemostat (Sixfors, HT) with oxygen and pH probes to monitor the culture. The aeration rate was set at 4.5 l/h and pH was kept at 7.2 +/- 0.1. For minimal glucose media, 40 mM MOPS media (M2120, Teknova) was utilized with glucose (0.4%, Sigma G8270), ammonia (9.5 mM NH4Cl, Sigma A9434) and phosphate (1.32 mM K2HPO4, Sigma P3786) added separately. For defined rich media, the minimal media is supplemented with 10x ACGU (M2103, Teknova) and 5X Supplement EZ (M2104, Teknova). For carbon- and nitrogen-limited media, glucose and ammonia concentrations were reduced by 5-fold (0.08% and 1.9mM respectively). Phosphorus-limited medium contains 0.132 mM $K_2HPO_4$. $\Delta leuB$ and $\Delta rplA$ mutants were produced by P1 transduction from the KEIO collection [77] into Escherichia coli strain NCM3722. Cell growth was monitored by checking absorbance at 600 nm using a spectrophotometer (GENESYS 20, Thermo Scientific).

### Experimental measurement of total RNA

The method for RNA measurement was adapted from You et al. [78]. The culture was 1.5 mL and centrifuged at 13,000g for 1 min to form pellets. The pellet was frozen on dry ice and the supernatant was used to measure absorbance for cell loss at 600 nm. Then the pellet was washed twice with 0.6 M $HClO_4$, digested with 0.3 M KOH at 37°C for 1h, and precipitated with 3 M $HClO_4$ to collect the supernatant. Then the pellets were extracted again with 0.5 M $HClO_4$. The supernatant was mixed and the absorbance was measured at 260 nm using Tecan Infinite 200 Pro (Tecan Trading AG, Switzerland). Finally, the total RNA concentration was the multiplication product of the absorbance value of A260 and the extinction coefficient (31 μg RNA mL$^{-1}$).

### Experimental measurement of total protein

The protein measurement method was adapted from You et al. [78]. The culture was 1.5 mL and centrifuged at 13,000g for 1 min to form pellets. The cells were washed with 1mL MOPS buffer once, suspended in 200 μL water again, and then placed on dry ice. All the supernatant was collected and cell loss was measured with A600nm. Then the samples were thawed to measure protein content. The samples were added with 100 μL 3M NaOH and heated at 98°C for 5

min. The samples were cooled to 20˚C for 5min. After that, 300 μL 0.1% $CuSO_4$ was added to the samples for biuret assay. The samples were incubated at room temperature for 5 min and centrifuged at 13,000g for 1 min. The supernatant was then collected and the absorbance of 200 μL sample volume was measured at 555 nm using software Gen5 in a Microplate reader (Synergy HT, BioTek). The total protein concentration in the cell was inferred using a known concentration of appropriately diluted albumin (23209, Thermo).

## Quantification of the total number and fraction of ribosomes

At 37˚C, 200 ml of cells were extracted from cultures and passed through cellulose acetate membranes (CA029025, Strelitech) with a 90 mm, 0.2 m-pore size. Then the cells were quickly frozen in liquid nitrogen after being scratched with a clean, previously warmed stainless-steel spatula. It took no more than two minutes to complete the filtration process in order to maintain the initial physiological state. Cell pellets and 650 μl of frozen nuggets of lysis buffer (20 mM Tris-HCl pH 8.0, 10 mM $MgCl_2$, 100 mM $NH_4Cl$, 0.4 percent Triton X-100, 0.1 percent NP-40, and 1 mM Chloramphenicol, 100 U ml$^{-1}$ RNase-free DNase I (04716728001 Roche)) were combined in a pre-chilled 10 ml jar (014620331, Retsch). The Cryomill (Retsch) was run at a 15 Hz pulverization rate for 15 minutes. RNA concentrations ranging from 80 g to 500 g in 200 μl of thawed cell lysates were measured using NanoDrop. Lysates were loaded to 10–55 percent linear sucrose gradients (20 mM Tris-HCl pH 8.0, 10 mM $MgCl_2$, 100 mM $NH_4Cl$, and 300 μM Chloramphenicol) made by GradientMaster for overall polysome quantification. The gradients were put in a SW41Ti bucket and centrifuged in a Beckman Coulter Optima XE-100 Ultracentrifuge for two hours at 4˚C and 35,000 rpm. Utilizing the BioComp Gradient Fractionator, the gradients were fractionated, and an ultraviolet monitor recorded the absorption curves at 254 nm (EM-1, BioRad). Polysome profiling was performed to quantify the ribosome fraction. The experimental methods were adapted from Li et al. [18]. The polysome profiling data was processed using customized MATLAB codes. The baseline absorbance was estimated using the average of the last 50 readings where RNA was not detected, and this background was subtracted. By fitting the exponential decay function to the first peak of the non-ribosomal signal source, free nucleotides and tRNA backgrounds were removed. Then each ribosome peak was selected and quantified by the area under the curve. The fraction of 70S and Polysomes were calculated from the area under the curve.

It is not possible to distinguish the free 70S peak and bound 70S peak because of the relatively small mass of mRNA compared to ribosome. To quantify the fraction of free 70S and bound 70S, we adapted the method from Li et al [18]. 100 mM $NH_4Cl$ was replaced with 170 mM KCl, and high potassium causes free 70S to shift to a lower density but does not shift the mRNA-bound 70S [79]. Cytolysis products were loaded into 10–30% linear gradient and centrifuged at 35,000 r.p.m. in a SW41Ti barrels for 5h at 4˚C. Then the MATLAB file-exchange scheme, Peakfit (2.0) esd used to fit the overlapped peaks (70S without mRNA, and 70S with mRNA binding) into two Gaussian distributions.

The total number of ribosomes was calculated based on the experimentally measured parameters as

$$R_t = V_c \cdot C_p \cdot RPR \cdot \frac{f_r}{m_r},$$

where the $V_c$ is cell volume ($m^3$) [80], $C_p$ is concentration of proteins ($g/m^3$) [81], $RPR$ represents RNA-to-protein ratio, $m_r$ is the mass of the rRNA component of a ribosome (g) [82], and $f_r$ is the fractional mass of rRNA among total RNA. The quantification of $f_r$ was adapted from Li et al. [18].

### *lacZ* induction and translational elongation rate measurement

The measurement of ribosome elongation rate was adapted from Zhu et al. [83]. Isopropyl-β-D-thiogalactoside (IPTG) (I2481C-25, Gold Biotechnology) with concentration of 5mM was added to the culture. Every 15 seconds, 1 mL of culture medium was taken and placed in a tube containing 10 μL of 100mm chloramphenicol, immediately frozen in liquid nitrogen and stored at -20°C, followed by subsequent measurements. After thawing, 400 μL of the sample was added to 100 μL of 5xZ buffer solution (0.3M $Na_2HPO_4.7H_2O$, 0.2M $NaH_2PO_4.H_2O$, 50mM KCl, 5mM $MgSO_4$, 20 mM β-mercaptoethanol) and incubated at 37°C for 10 minutes. 100 μL 4 mg mL$^{-1}$ 4-methylumbelliferyl-β-D-galactopyranoside (MUG, 337210010, ACROS Organics) in DMSO was added to each sample every 10 s for precise control of the reaction time. The samples were incubated in Eppendorf Thermomixer R at 37°C at a mixing rate of 1400 r.p.m. for 30 min to 2 h, according to the enzyme expression levels. Then we added 300 μL 1 M $Na_2CO_3$ to stop the reaction. The tube was spun down at 16,000g for 3 min to precipitate cell debris. Finally, the fluorescence of 200 μL supernatant was measured with a microplate reader (365 nm excitation and 450 nm emission filter). We integrated the signals and performed a linear fit to infer the ribosome elongation rate. According to the previous study [83], the elongation time was corrected by subtracting 10 s from the measured delay time.

### RNA extraction and ribosome profiling

The method of RNA extraction and ribosome profiling is described in Li et al. [18]. The cell collection step was the same as for polysome profiling in Li et al. [18] except that 1mM chloramphenicol was utilized in the sucrose solution. The footprinting and library preparation steps were adapted from Li et al. [17] After quantification of RNA concentration with NanoDrop, samples with 500μg RNA were digested with 750U MNase (10107921001, Roche) for 1 hour at 25°C before being quenched with 6mM EGTA. The lysates were then layered onto a 10%-55% sucrose gradient and centrifuged. The monosome fraction was collected and snap frozen in liquid nitrogen. There were no observed polysome peaks, which indicated thorough digestion. The RNA was separated using hot phenol and size selected on 15% TBE-Urea PAGE gels run for 1 hour at 210V. Gels were stained with SYBR Gold and visualized using Dark Reader (Clare Chemical Research). Finally, RNA fragments with size between 25–40 nt were extracted using isopropanol precipitation.

### Library preparation and sequencing

RNA fragments from footprints were dephosphorylated at the 3' end by PNK (M0201, NEB). The repaired fragments were linked to the Universal miRNA Cloning Linker (S1315S, NEB), reverse transcribed (18080044, Thermo), and circularized (CL4111K, Epicentre). rRNA was subtracted from the circularized samples before PCR amplification (M0531L, NEB) and size selection. High quality PCR samples were checked by Bioanalyzer highly sensitive DNA chip. Deep sequencing was performed by Illumina HiSeq 2500 on Rapid flowcells with settings of single end and 75 nt-long read length.

### Mapping and sequencing data analysis

Data processing including barcode splitting, linker trimming, and mapping were performed using Galaxy [84]. The processed reads were mapped to Escherichia coli genome escherichia_-coli_k12_nc_000913_3 from the NCBI database with the BWA short read mapping algorithm [85]. Only the reads between 20–45 nt that aligned to the coding region were extracted for further analysis.

To infer the ribosome A-site position, python package Plastid [86] was used to align the 3' end of reads to the stop and start codons [87], which are known to have higher ribosome densities. We found that the offsets were 12 nt for stop codon and 15 nt for start codon. Therefore, we utilized 11nt for A site position and 14nt for P site. Further analysis was done using MATLAB and R codes.

## Analysis of deep-sequencing data

The counts from ribosome profiling and RNA-seq were used to calculate relative translation efficiency (RTE) for each transcript:

$$[RTE] = \frac{[footprinting\ counts]/[gene\ length]}{[relative\ mRNA\ level]},$$

where the footprinting counts were normalized by the total counts in one experiment, reflecting the percentage of ribosomes occupied by a gene. The ratio of footprinting counts to gene length reflects the relative ribosome density: the percentage of ribosomes occupied by per unit length of a gene. The relative mRNA levels were also normalized to the total counts and gene length as reads per kilobase million (RPKM). In addition, genes with $\log_{10}$(RNA-seq RPKM) > 1.5 were selected for subsequent analysis (selected genes n = 2914).

Mean levels were taken as the average of the 12 conditions for analyzing the correlation between the mRNA level and RTE across genes. The Spearman's rank correlation coefficient was used for correlations both across genes and across conditions. In order to test the significance of the distribution of correlation coefficients between mRNA and RTE across conditions, the RTE values for each gene were randomly scrambled among the 12 conditions. The resulting randomly ordered RTEs were used to recalculate the distribution of correlation coefficients, which was considered as the null distribution. Then we used the package *kstest2* in MATLAB to test whether the two distributions are significantly different, and calculated the *p*-value.

When comparing two different nutritional restriction conditions, the RNA-seq RPKM were averaged for three biological replicates. Then we screened for differential gene groups with $\log_2$(mRNA fold change) > 4 or < -4 and *p*-value < 0.05. To test the significance of RTE fold changes for the genes with differentially expressed mRNA, we first calculate the RTE fold change distribution for this group of genes. Then the distribution of the RTE fold changes for the whole set of 2914 genes was considered as the null distribution. A *p*-value was calculated using student's *t*-test for the two distributions. All the above processes were performed with Matlab2020a.

## GO analysis and KEGG pathway analysis

Functional enrichment analysis was carried out using function *enrichGO* in R package *clusterProfiler* [88]. In addition, genome wide annotation *org.EcK12.eg.db* for *E. coli* strain K12 was used. The enrichment results were filtered with an adjusted *p*-value < 0.05. Furthermore, function *dropGO* was used to refine gene ontology level. Besides, KEGG pathway enrichment analysis was carried out using function *enrichKEGG* in R package *clusterProfiler* [88]. Genes contained in the 82 pathways of *E. coli* strain K-12 MG1655 were obtained from https://www.genome.jp/kegg/pathway.html.

## Codon usage analysis

The codon frequency of a gene was defined as the ratio of the number of a certain codon to the total number of codons. The frequencies of 64 codons constituted the codon frequency vector

of a gene. Then we calculated the background codon frequencies from the complete set of analyzed genes. To characterize the bias for a gene towards certain codons, the background codon frequencies were subtracted from the codon frequency vector.

Before comparing the codon usage between different pathways, the overlapped genes were removed. Then we calculated the average codon frequencies for all genes in a pathway. As shown in Fig 6A, the rarity of codons was ranked according to their background frequencies.

### Evaluation indices for RTE variability

We used three different evaluation indices: the variance, the Fano factor, and the coefficient of variation (CV). The variance is defined as

$$\text{var}(\text{RTE}) = \frac{\sum (\text{TE} - \overline{\text{RTE}})^2}{n - 1},$$

where $\overline{\text{RTE}}$ is the sample mean of RTE, and the $n$ is the sample size of RTE. The Fano factor is defined as

$$\text{Fano}(\text{RTE}) = \frac{\sigma^2_{\text{RTE}}}{\mu_{\text{RTE}}},$$

where $\sigma^2_{\text{RTE}}$ is the variance of RTE, and the $\mu_{\text{RTE}}$ is the sample mean of RTE. The CV is defined as

$$\text{CV}(\text{RTE}) = \frac{\sigma_{\text{RTE}}}{\mu_{\text{RTE}}},$$

Where $\sigma_{\text{RTE}}$ is the standard deviation of RTE, and the $\mu_{\text{RTE}}$ is the sample mean of RTE.

### Random forest algorithm

We used the package *TreeBagger* in MATLAB to build the binary classification model. The number of trees was set to 200 and the minimum number of observations per tree leaf was set to 5. The number of variables to select at random for each decision split was set to the square root of the total variable number. In our model, the total variable number is 64, corresponding to the 64 codons. Finally, Breiman's random forest algorithm was invoked to perform the training [89].

As stated in the main text, features such as frequencies of the 64 codons, mRNA level, RTE absolute value, the distribution of the third base of codons, and gene length were selected and combined to determine their contribution to classification results. In addition, the frequencies of typical translation pause motifs were also used as classification features.

1000 random samplings of the dataset were performed to exclude the contingency of results. We used true positive rate to evaluate the sensitivity, defined as

$$\text{sensitivity} = \frac{\text{TP}}{\text{TP} + \text{FN}},$$

where TP and FN refer to the number of true positives and false negatives, respectively. The specificity is defined as

$$\text{specificity} = \frac{\text{TN}}{\text{TN} + \text{FP}},$$

The area under curve (AUC) was calculated as the area under ROC curve. To calculate the sensitivity and specificity, a classification threshold is needed. The score for each gene from

the model is in the range of [0, 1]. If the score is above the threshold, it is considered a positive sample, otherwise it is considered a negative sample. The results shown in Table 2 used 0.5 as the classification threshold.

## Supporting information

**S1 Text. Text for the theoretical analysis for the distribution of correlation coefficients between mRNA level and randomly scrambled RTE, quantify the contribution of codon usage to two different definitions of RTE variability, and supplementary figures and tables corresponding to the main text.**
(PDF)

**S1 Data. mRNA levels normalized by RPKM from RNA-Seq.**
(XLSX)

**S2 Data. Ribosome footprintings normalized by RPKM from ribosome profiling.**
(XLSX)

## Author Contributions

**Conceptualization:** Sophia Hsin-Jung Li, Ned S. Wingreen, Zemer Gitai, Zhiyuan Li.

**Data curation:** Di Zhang, Sophia Hsin-Jung Li, Christopher G. King.

**Formal analysis:** Di Zhang.

**Funding acquisition:** Ned S. Wingreen, Zemer Gitai, Zhiyuan Li.

**Investigation:** Christopher G. King, Ned S. Wingreen, Zemer Gitai.

**Methodology:** Di Zhang.

**Project administration:** Zhiyuan Li.

**Resources:** Sophia Hsin-Jung Li.

**Validation:** Sophia Hsin-Jung Li.

**Visualization:** Di Zhang.

**Writing – original draft:** Di Zhang.

**Writing – review & editing:** Ned S. Wingreen, Zemer Gitai, Zhiyuan Li.

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
