## [Decision Letter · Decision Letter 0]

27 Jun 2022

Dear Dr. Li,

Thank you very much for submitting your manuscript "Global and gene-specific translational regulation in Escherichia coli across different conditions" for consideration at PLOS Computational Biology.

As with all papers reviewed by the journal, your manuscript was reviewed by members of the editorial board and by several independent reviewers. In light of the reviews (below this email), we would like to invite the resubmission of a significantly-revised version that takes into account the reviewers' comments.

While all three reviewers are positive about the work, they raise several technical points that need to be addressed. Please pay special attention to the concerns raised by Reviewer 2 about normalization and about better highlighting the novelty of the work and the main message of the work.

We cannot make any decision about publication until we have seen the revised manuscript and your response to the reviewers' comments. Your revised manuscript is also likely to be sent to reviewers for further evaluation.

Sincerely,

Saurabh Sinha

Guest Editor

PLOS Computational Biology

Ilya Ioshikhes

Deputy Editor

PLOS Computational Biology

While all three reviewers are positive about the work, they raise several technical points that need to be addressed. Please pay special attention to the concerns raised by Reviewer 2 about normalization and about better highlighting the novelty of the work and the main message of the work.

Reviewer's Responses to Questions

**Comments to the Authors:**

Reviewer #1: The correlation between transcription output and translation output remains controversial but is of great importance and broad impact in molecular biology and systems biology (including the application of synthetic biology). Combing RNA-seq and Ribo-seq, Zhang et al use E. coli as a model system and quantify the mRNA level and translation efficiency (TE) for individual genes in various poor nutrients.

As pointed out by Gene Wei Li (Curr Opin in Microbiol, 2015, 23:66-71) (the author could cite this paper), the major determinant of TE is still largely unknown, e.g., RBS (SD sequence) has been found to be not quite related to TE. Therefore, the issue of TE regulation is a very important open direction to investigate. This paper provides very useful large-scale TE information of bacteria to guide future theoretical quantitative studies and synthetic biology applications.

The paper is generally well written and is of broad interest for the community of Plos Comput Biol. I have just a few minor comments for the authors, mostly for clarification.

1. Line 169: The definition of RTE: the ratio of the transcript abundance of the ribosome profiling to relative mRNA level. I suggest making it more clear here: first introduce the definition of TE: the rate of protein production per mRNA (the translating ribosome number per mRNA). Then claim that here TE is represented by dividing rate of synthesis of each protein (ribosome density by ribosome profiling) by the corresponding mRNA levels as measured by RNA-seq, namely RTE.

2. Figure 2C: please provide the definition of actual RTE and random RTE. Not quite clear here.

3. In Figure 2D and 2E (also such as Figure 3C, 3D and related places), I suggest also showing the data of ribosome density (from ribosome profiling) at the middle between mRNA and RTE. The ribosome density is the total output of protein production rate of a specific gene. You have RTE=ribosome density/mRNA, correct?

4. About the point of rare codon issue, according to Li et al 2012 Nature, codons usage seems not impact in vivo translation rates (in rich medium). Does the author note sth different in poor nutrient conditions? If so, this could be another useful information to show and make some useful discussions.

Reviewer #2: Zhang et al attempts to uncover the gene-specific regulation at the level of transcription and translation in response to carbon, nitrogen and phosphate limitations. It is a timely question, the authors try to tackle different points and probably it is not clear (at least to me) what the main message of this work should be. The main result that has been highlighted is that “random forest model suggests that codon usage partially predicts a gene’s cross-condition variability in translation efficiency”. However, this is just a small part of the manuscript, which arrives at the end and that could be better discussed/integrated.

1. The introduction is rather well written, however I believe that there is some important literature missing. For instance, when the authors discuss how “E. coli cells are known to cope with different nutrient limitations by changing ribosomal synthesis and usage strategies” (line 78), they should probably cite the pioneering works by Schaechter or Bremer&Dennis, or more recent publications (e.g. PMID: 21097934); there are several “theoretical models … developed to provide an integrative picture of translational regulation”, other than the cited ones (e.g. PMID: 24688632, 29694095, 27591251, 29448386). The literature about the role of secondary structures on the efficiency of translation is vaste (e.g. 31760691), the authors could probably explore this field a bit more in depth.

2. One of the main questions is the relation between this manuscript and other recent works from other groups (for instance Mori et al. PMID: 34032011), where data is available also for a broader range of growth. Can the authors clarify the novelty of their work?

3. Methods are often taken from a previous publication of the group (ref 18). However, it is useful to explain how the measurements are done and when the values are not directly experimentally measured and when are inferred by fitting the model (e.g. elongation rates).

4. Line 132: The free ribosome pools decreased as growth rate increased… how is that measured? To my understanding there is no experimental way to determine the amount of free (or bound) ribosomes. It is possible to use polysome profiles (as more or less said in the caption of Figure 1) but the way the profiles can be analysed remains arbitrary (70S are considered free or bound? What about 30S and 50S?

5. Panels C-F of figure 1 are difficult to read. At least the authors should give the range and draw a few axis ticks. Labels in panel A can be improved…

6. Why do the authors not investigate TE and the fraction of transcript of the gene of interest with respect to the total amount of mRNA? I imagine that this would be more meaningful, and probably one can avoid possible normalisation problems (PMID: 32284352). I may be missing something, but it seems to me a more robust analysis to understand the tradeoff of transcription (mRNA) and translation (RTE).

7. Figure 2: “Mean levels were taken as the average of all 12 conditions (Table 1)”. Why? Would not be more useful to do the same for each condition (or for at least each growth rate).

8. Figure 2D-E: Scatter plots would be more useful (and one can color the points with the growth rate and conditions with different symbols).

9. Line 278: “almost all of their RTE…” Can the authors provide a percentage?

10. About the upregulation of RTE and mRNA, I am slightly worried that this results could be an artifact of normalisations of the data (the more mRNAs one has, more ribosome footprints etc etc). Can the authors comment on that and check that this is not the case?

11. It is somehow difficult to understand the main message of Figure 5. The authors should try to clarify it in the main text (and maybe chose a different way to show the data?). The same it is also true for panels G and H of the same figure: it is not clear why the authors decide to talk about the codon usage at this point (and not wait till the following section).

12. Figure 6A. Other than the random forest, the authors could randomly choose 200 lowly expressed genes and 200 highly expressed genes (without looking at RTE) and try to do the same plot.

Reviewer #3: In this work, the authors analyzed the translational regulation which might be a major cause that led to the well-known discrepancy between mRNA and protein levels. They first showed that E.coli cells achieved the same growth rate at different nutrient conditions by differentially regulate three ribosome features. The authors then combined ribosome profiling and RNA-seq to quantify the relative translation efficiency (RTE). By analyzing the pattern and correlation of mRNA and RTE levels, the author assessed the gene-specific and condition-specific translational regulation. They further demonstrated the relationship between codon frequencies and translational regulation patterns. These results would be important to help people understand the translational regulation of genes and help bridge the gap between mRNA and protein. Here are some comments and revisions that might need to be made:

1. In figure 1A, data points from the same condition but at different growth rate are connected by straight lines, which seem to suggest linear relationship between the mRNA-protein ratio and the growth rate. However, with only two different growth rate levels, there are not enough data points to make assumptions on the linearity.

2. The colors representing free 70S monosomes, mRNA-bound 70S monosomes and mRNA-bound polysomes are not correctly noted in the figure legends of figure 2B. For example, free 70S is represented in dark color in figure 2B. But the figure legends says that it is represented in white.

3. A positive correlation of coefficient of determination of 0.3 between mean mRNA and RTE was observed when cross-gene relationship was analyzed. However, more negatively correlated mRNA and RTE pairs were identified when the cross-condition correlation analysis was performed. Is there any explanation for the conflicted observations in the cross-gene and cross-condition correlation analysis?

4. It would be helpful if some statistics (like a p-value) could be provided to show the skewness of the distribution of the Spearman’s correlation of 2914 genes. (figure 2C)

5. The fold enrichment of the top 300 genes with the most negative correlation are generally weaker than the fold enrichment of the top genes with the most positive correlation (figure 3A and 3B). What could be the reason for this?

6. Authors claimed that “pathways with similar translational regulation patterns tend to share similar codon usage” by comparing the codon frequencies of two pathways that have similar RTE variance across conditions or their mean RTEs were correlated. It would be helpful to see some control analysis: 1) How similar the codon usage would be for two randomly selected pathways? 2) Are there two pathways with related biological functionals but very different RTE variance pattern? 3) If there exist such pathways, do they share similar codon frequencies?

**Have the authors made all data and (if applicable) computational code underlying the findings in their manuscript fully available?**

Reviewer #1: Yes

Reviewer #2: **No: **Scripts for data analysis/plotting figures are not available. Although raw data are made available (GEO) it would have been useful to publish processed data (RTE, mRNA,...)

Reviewer #3: Yes

PLOS authors have the option to publish the peer review history of their article (what does this mean?). If published, this will include your full peer review and any attached files.

Reviewer #1: No

Reviewer #2: No
---

## [Decision Letter · Decision Letter 1]

6 Oct 2022

Dear Dr. Li,

We are pleased to inform you that your manuscript 'Global and gene-specific translational regulation in Escherichia coli across different conditions' has been provisionally accepted for publication in PLOS Computational Biology.

Best regards,

Saurabh Sinha

Guest Editor

PLOS Computational Biology

Ilya Ioshikhes

Section Editor

PLOS Computational Biology

Editor's note: Two of the reviewers are fully satisfied with the revision, while Reviewer 2 suggests a clarification and a notation change. The authors may choose to incorporate those suggestions into their final manuscript submission.

Reviewer's Responses to Questions

**Comments to the Authors:**

Reviewer #1: The authors has addressed my concerns and I am happy to recommend its publication.

Reviewer #2: With this revision, Zhang et al have improved the manuscript and answered my previous comments. I still have a couple of points related to the equation they used to compute the total amount of ribosomes and about the RTE.

1. At line 658, the authors give the equation to compute the total amount of ribosomes: R_t = V_c C_\\rho RPR f_r / m_r. I am confused and I might be missing something trivial, but I would have said that the total number of ribosomes can be computed, for instance, from Nr = 0.86 R / m_r (where R is the total mass of RNA and 0.86 is the fraction of rRNA), or equivalently from N_r = total mass of ribosomal proteins / mass of ribosomal proteins in a single ribosome. If I understand correctly the product V_c C_\\rho RPR is simply equal to R (the total mass of RNA), and I do not understand why the authors do not simply write Nr = 0.86 R / m_r (with their notations).

2. The authors refer to the TE or RTE as a rate. However, this quantities are related to the ribosome density (it is also mentioned in the manuscript) and it carries no information about the timescales (i.e., it is not a rate). I acknowledge that in the literature people always consider TE as a production rate. However, I would encourage the authors to stop committing this mistake and remove all the relations between TE and rates in the text.

Also, as it is a density, and since the ribosome density has an upper bound (it cannot exceed 1 ribosome/30 nt), could the authors check whether their values of RTE are consistent with that bound?

Reviewer #3: The authors have carefully addressed the comments.

**Have the authors made all data and (if applicable) computational code underlying the findings in their manuscript fully available?**

Reviewer #1: Yes

Reviewer #2: Yes

Reviewer #3: Yes

PLOS authors have the option to publish the peer review history of their article (what does this mean?). If published, this will include your full peer review and any attached files.

Reviewer #1: No

Reviewer #2: No

Reviewer #3: No

---

## [Editor Report · Acceptance letter]

16 Oct 2022

PCOMPBIOL-D-22-00654R1 

Global and gene-specific translational regulation in *Escherichia coli* across different conditions

Dear Dr Li,

I am pleased to inform you that your manuscript has been formally accepted for publication in PLOS Computational Biology. Your manuscript is now with our production department and you will be notified of the publication date in due course.

With kind regards,

Zsofia Freund
